# What happens to patient experience when you want to see a doctor and you get to speak to a nurse? Observational study using data from the English General Practice Patient Survey

Charlotte A M Paddison,[1] Gary A Abel,[2] Jenni Burt,[3] John L Campbell,[2] Marc N Elliott,[4] Valerie Lattimer,[5] Martin Roland[3]

[1]Nuffield Trust, London, UK
[2]University of Exeter Medical School, Exeter, UK
[3]Cambridge Centre for Health Services Research, Primary Care Unit, Institute of Public Health, School of Clinical Medicine, University of Cambridge, Cambridge, UK
[4]RAND Corporation, Santa Monica, California, USA
[5]School of Health Sciences, Norwich Research Park, University of East Anglia, Norwich, UK

**Correspondence to**
Dr Charlotte A M Paddison;
charlotte.paddison@nuffieldtrust.org.uk

## ABSTRACT

**Objectives** To examine patient consultation preferences for seeing or speaking to a general practitioner (GP) or nurse; to estimate associations between patient-reported experiences and the type of consultation patients actually received (phone or face-to-face, GP or nurse).

**Design** Secondary analysis of data from the 2013 to 2014 General Practice Patient Survey.

**Setting and participants** 870 085 patients from 8005 English general practices.

**Outcomes** Patient ratings of communication and 'trust and confidence' with the clinician they saw.

**Results** 77.7% of patients reported wanting to see or speak to a GP, while 14.5% reported asking to see or speak to a nurse the last time they tried to make an appointment (weighted percentages). Being unable to see or speak to the practitioner type of the patients' choice was associated with lower ratings of trust and confidence and patient-rated communication. Smaller differences were found if patients wanted a face-to-face consultation and received a phone consultation instead. The greatest difference was for patients who asked to see a GP and instead spoke to a nurse for whom the adjusted mean difference in confidence and trust compared with those who wanted to see a nurse and did see a nurse was −15.8 points (95% CI −17.6 to −14.0) for confidence and trust in the practitioner and −10.5 points (95% CI −11.7 to −9.3) for net communication score, both on a 0–100 scale.

**Conclusions** Patients' evaluation of their care is worse if they do not receive the type of consultation they expect, especially if they prefer a doctor but are unable to see one. New models of care should consider the potential unintended consequences for patient experience of the widespread introduction of multidisciplinary teams in general practice.

## INTRODUCTION

Patient experience is a core dimension of healthcare quality[1] and measuring patient experience enables the views of service users to be incorporated into the evaluation and improvement of health services at

### Strengths and limitations of this study

► Large national survey with respondents including 870 085 patients from 8005 English general practices.
► Appointment choice (mode and type of practitioner), and patient-reported experience, all collected in a single survey.
► Thirty-four per cent response rate, though typical for similar national surveys.
► Patient-reported confidence and trust assessed using a single item within the General Practice Patient Survey.

local and national level.[2 3] In the UK, 90% of all National Health Service (NHS) patient contacts occur in primary care. Though primary care is highly valued by the UK public, the primary care workforce is under unprecedented strain[4 5] and clear policies for reforming primary care in England are now emerging.[6 7] These include the development of wider multidisciplinary teams that make greater use of nurses, physicians' associates, pharmacists and administrative support. It is anticipated that an expanded workforce will free up clinical time for doctors, enabling them to devote more time to the delivery of care for complex patients and spend less time on administration and minor health issues.[7]

At present it is unclear what impact these proposed changes to the primary care workforce might have on patient choice and quality of care, including patient experience. Multidisciplinary teams could increase patient choice by improving access to nurses and pharmacists, but their introduction could also have unintended adverse impacts, for example, on patients who were no longer able to see the type of clinician of their choice.

Given the policy emphasis on promoting patient choice, evidence is needed on the acceptability to patients of seeing different practitioners, especially when their original intention had been to see a general practitioner (GP). Systematic reviews suggest that nurse–GP substitution in primary care can produce similar or better quality of care and high levels of patient satisfaction,[8 9] but there is also evidence that some patients do not want nurses to substitute for GPs.[10] Relational continuity of care, together with perceived trust and confidence, have emerged as important factors in understanding why some patients may prefer to see a GP and, more specifically, a GP who knows them well.[11]

In this primary care-based study, we examine patient consultation preferences and their association with patient-reported experiences, using data from the English General Practice Patient Survey (GPPS) to address the following questions:

1. To what extent do patients in primary care request an appointment with a doctor or a nurse, and how often is this request met?
2. When patients are unable to get an appointment of their choice (phone or face-to-face, GP or nurse), is this associated with poorer patient-reported communication or lower trust and confidence?

## METHODS

We analysed data from the 2013–2014 GPPS (http://www.gp-patient.co.uk), which is sent as a mail questionnaire to 2.6 million patients in England each year.[12] We estimated preference for nurse and GP appointments from the question which asked respondents 'Last time you wanted to see or speak to a GP or nurse from your GP surgery: what did you want to do?' with response options coded into five categories (see online supplementary appendix table 1):

▶ See or speak to a GP.
▶ See or speak to a nurse.
▶ A home visit
▶ Was not sure or did not mind.
▶ Wanted more than one of the previous four categories.

Practice population estimates were computed using survey weights that account for survey design and non-response.[13] Variation in preferences for GP and nurse appointments were explored using crude and adjusted logistic regression using a binary classification (wanting to see or speak to a nurse vs wanting to see or speak to a GP as the outcome—see online supplementary appendix table 2 for detail of how categories were constructed), excluding those indifferent as to whether they saw a doctor or a nurse. Fixed effects included in the models were patient-reported age, gender, ethnicity, confidence in managing their own health and presence of a long-term condition, as well as a measure of social deprivation based on the patient's postal code of residence (national quintiles as categories). General practice was included as a random effect (intercept). All analyses were restricted

to patients with complete information on all variables included in the adjusted model.

Next, we investigated whether people who did not get an appointment with the type of practitioner of their choice were more likely to report poorer patient experience. For simplicity this analysis was restricted to those who only endorsed one response option each for the question on what type of appointment they had requested and for the question on what they got. Sixteen categories, combinations of seeing or speaking to a health professional and whether the appointment was with a GP or a nurse, were created. The dataset was then split into those whose last appointment was with a GP and those whose last appointment was with a nurse, and these were modelled separately.

Two series of regression models were run. The first series of models used responses to the question 'Did you have confidence and trust in the GP (nurse) you saw or spoke to?' as the outcome. Response options were: 'yes, definitely'; 'yes, to some extent' and 'no, not at all' which were scored as 100, 50 and 0, respectively. Crude differences between the categories were estimated for trust and confidence in GPs and, separately, for nurses, using linear regression. Mixed-effect linear regression was then used for an adjusted analysis, including age, gender, ethnicity, confidence in managing their own health, the presence of a long-standing heath condition and deprivation. Practice was included as a random effect (intercept), and analyses were restricted to those who had complete information on all sociodemographic and health variables in this study.

A second series of models using the same approach (linear regressions to estimate crude differences between categories, and mixed-effect linear regression for analyses adjusting for health and demographic characteristics and additionally controlling for practice) were then run using composite patient-rated communication scores as the outcome. This composite score was the mean of all responses to five questions on patient-reported experiences of communication for patients who answered at least three of these questions. Scores were computed separately for doctor communication (five items) and nurse communication (five items). The questions asked how good the GP or nurse was at 'giving you enough time', 'listening to you' or 'explaining tests and treatments', 'involving you in decisions about your care' and 'treating you with care and concern'. As in our previous work, responses to these patient experience items were linearly rescaled to give a score from 0 to 100 in order to facilitate comparison between items.[14 15]

## RESULTS

Among 870 085 primary care patients from 8005 practices in England, more than three-quarters reported that they only asked to see or speak to a GP, while just under 15% only asked to see or speak to a nurse (table 1) with the remainder wanting a home visit, expressing no preference

**Table 1** Descriptive statistics showing what patients wanted to do last time they contacted their GP surgery, and weighted percentages showing the association between the type of appointment patients wanted and the appointment they got

| Last time you wanted to see or speak to a GP or nurse from your GP surgery: what did you want to do? | N (weighted %)† | What type of appointment did you get?* | | | | |
| --- | --- | --- | --- | --- | --- | --- |
| | | To see and/ or speak to a GP, % | To see and/ or speak to a nurse, % | A home visit, % | One or more of the above, % | Total, % |
| See and/or speak to a GP | 653 526 (77.7) | 95.9 | 2.9 | 0.1 | 1.1 | 100.0 |
| See and/or speak to a nurse | 139 300 (14.5) | 6.3 | 92.2 | 0.1 | 1.4 | 100.0 |
| A home visit | 12 873 (1.2) | 15.7 | 2.2 | 79.6 | 2.5 | 100.0 |
| Did not mind/was not sure | 15 404 (2.4) | 67.1 | 29.1 | 1.1 | 2.7 | 100.0 |
| One or more of the above | 48 982 (4.3) | 29.7 | 8.6 | 0.7 | 61.1 | 100.0 |
| Total | 870 085 (100.0) | 77.6 | 17.5 | 1.1 | 3.8 | 100.0 |

*Of those who answered the question 'Last time you wanted to see or speak to a GP or nurse from your GP surgery: what did you want to do?' (n=870 085) only 754 551 also answered the question 'What type of appointment did you get?'.
†Weighted percentages are calculated using survey design and non-response weights (by age, gender, geographical location and GP practice, full details Technical Annex GP Patient Survey[13]).
GP, general practitioner.

or asking for multiple types of appointment. Comparing the type of appointment patients wanted and the type of appointment they actually received, we found that the great majority of people got what they wanted. As shown in table 1, 96% of people who wanted to see or speak to a GP achieved this, compared with 92% of people who wanted to see or speak to a nurse. The number wanting a home visit who got one was lower at 80%.

There were 685 244 respondents who only wanted to see or speak to either a nurse or doctor who also had complete information for all covariates of interest. Of these 122 597 (weighted percentage 16.0%) wanted to see or speak to a nurse, with the remainder wanting to see or speak to a GP. Variation in this percentage is shown in table 2 along with crude and adjusted ORs. Older patients were less likely to want to see or speak to a GP than younger patients (eg, ≥85 vs 55–64 years, adjusted OR 0.76, 95% CI 0.74 to 0.79). Ethnic minority group patients were much more likely to want to see or speak to a GP (eg, Asian vs White patients, adjusted OR 1.51, 95% CI 1.47 to 1.56). Similarly those who were not confident in managing their own health were more likely to want to see or speak to a GP (eg, Not at all confident vs Very confident, adjusted OR 1.73, 95% CI 1.61 to 1.85).

Being unable to get an appointment of the patients' choice was associated with differences in trust and confidence (table 3) and communication (table 4), among patients who reported seeing or speaking to a nurse or a GP in their last primary care consultation. The mean trust and confidence score for those who had asked to see a nurse and did so was 91.0 (on a scale of 0–100). For all other combinations patient-rated trust and confidence scores were, on average, lower (P<0.001). Patient-reported trust and confidence was lowest for those who wished to see a GP or wished to speak to a GP, and instead spoke to a nurse (adjusted mean difference compared with those who wanted to see a nurse and did see a nurse=−15.8, 95% CI −17.6 to −14.0 and −13.5, 95% CI −15.9 to −11.0,

respectively), followed by those who wanted to see a GP but ended up speaking to, rather than seeing, a GP (−6.5, 95% CI −7.0 to −5.9). Similar differences were found for patient-reported communication scores, with the lowest scores for those who wanted to see a GP but spoke to a nurse (adjusted mean difference compared with those who wanted to see a nurse and did see a nurse=−10.5, 95% CI −11.7 to −9.3).

## DISCUSSION

In a study of 870 085 respondents from general practices in England, we show that the majority of patients in primary care are able to see or speak to the health professional type (GP vs nurse) of their choice. However, a substantive minority—between 4% and 8%—are not. We found evidence that patient consultation preferences vary across patient groups. Patients with low confidence in managing their own health, younger patients and ethnic minority groups were more likely to want to see a GP.

When patients were unable to obtain the type of appointment of their choice, this was associated with lower confidence and trust in the health professional they saw, and poorer patient-rated communication. This difference was particularly large among patients who wished to see a GP and instead spoke to a nurse. A difference, though smaller in magnitude, was also observed for patients who wanted to see a nurse and got to speak to a GP. Patients whose appointments were with GPs reported similar experiences regardless of their initial preference for a nurse versus a doctor. Patient experience scores were also lower when the patient had asked to see a health professional (either a GP or a nurse), and ended up having a telephone call.

Our study, using data from a large national survey of patient experience in primary care, builds on what is known from previous research examining the acceptability to patients' of nurse–GP substitution[8 9 16 17] and

**Table 2** Associations between patient characteristics and wanting an appointment to see or speak to a GP

| | Wanted GP appointment N (%)*† | Wanted nurse appointment N (%)* | Crude OR Wanted GP appointment (95% CI)‡ | Adjusted OR Wanted GP appointment (95% CI)§ |
|---|---|---|---|---|
| Age | | | | |
| 18–24 | 23 978 (86.5) | 3953 (13.5) | 1.34 (1.29 to 1.39) | 1.29 (1.24 to 1.34) |
| 25–34 | 55 900 (86.1) | 9828 (13.9) | 1.26 (1.22 to 1.29) | 1.20 (1.17 to 1.23) |
| 35–44 | 78 731 (87.0) | 12 575 (13.0) | 1.38 (1.35 to 1.41) | 1.32 (1.29 to 1.35) |
| 45–54 | 106 693 (85.2) | 19 007 (14.8) | 1.24 (1.21 to 1.27) | 1.22 (1.20 to 1.24) |
| 55–64 | 117 426 (82.3) | 25 927 (17.7) | Reference | Reference |
| 65–74 | 107 427 (77.9) | 31 381 (22.1) | 0.76 (0.74 to 0.77) | 0.77 (0.76 to 0.79) |
| 75–84 | 57 635 (79.0) | 15 802 (21.0) | 0.81 (0.79 to 0.82) | 0.80 (0.79 to 0.82) |
| 85 or over | 14 857 (78.6) | 4124 (21.4) | 0.80 (0.77 to 0.83) | 0.76 (0.74 to 0.79) |
| Gender | | | | |
| Male | 246 540 (85.2) | 53 712 (14.8) | Reference | Reference |
| Female | 316 107 (82.8) | 68 885 (17.2) | 1.00 (0.99 to 1.01) | 0.99 (0.97 to 1.00) |
| Ethnicity | | | | |
| White | 495 207 (83.4) | 113 381 (16.6) | Reference | Reference |
| Mixed | 4432 (87.0) | 686 (13.0) | 1.48 (1.36 to 1.60) | 1.23 (1.14 to 1.33) |
| Asian | 33 748 (89.1) | 4303 (10.9) | 1.80 (1.74 to 1.85) | 1.51 (1.47 to 1.56) |
| Black | 14 442 (87.8) | 2151 (12.2) | 1.54 (1.47 to 1.61) | 1.30 (1.25 to 1.36) |
| Other | 14 818 (88.4) | 2076 (11.6) | 1.63 (1.56 to 1.71) | 1.41 (1.35 to 1.47) |
| Index of Multiple Deprivation group | | | | |
| 1—least deprived | 111 565 (83.3) | 25 422 (16.7) | Reference | Reference |
| 2 | 116 118 (83.1) | 26 848 (16.9) | 0.99 (0.97 to 1.00) | 0.98 (0.97 to 1.00) |
| 3 | 115 657 (83.5) | 26 334 (16.5) | 1.00 (0.98 to 1.02) | 0.97 (0.96 to 0.99) |
| 4 | 110 059 (84.6) | 22 543 (15.4) | 1.11 (1.09 to 1.13) | 1.03 (1.01 to 1.05) |
| 5—most deprived | 109 248 (85.5) | 21 450 (14.5) | 1.16 (1.14 to 1.18) | 1.03 (1.01 to 1.05) |
| Confidence in managing health | | | | |
| Very confident | 240 196 (82.3) | 58 806 (17.7) | Reference | Reference |
| Fairly confident | 283 776 (84.9) | 58 435 (15.1) | 1.19 (1.17 to 1.20) | 1.21 (1.19 to 1.22) |
| Not very confident | 33 166 (89.1) | 4562 (10.9) | 1.78 (1.72 to 1.84) | 1.75 (1.70 to 1.80) |
| Not at all confident | 5509 (88.2) | 794 (11.8) | 1.70 (1.58 to 1.83) | 1.73 (1.61 to 1.85) |
| Long-term condition | | | | |
| No | 216 594 (85.2) | 42 250 (14.8) | Reference | Reference |
| Yes | 346 053 (83.0) | 80 347 (17.0) | 0.84 (0.83 to 0.85) | 0.97 (0.96 to 0.99) |

ORs >1 indicate patient group more likely to want to see a GP than the reference group. Only those with complete information for all covariates are included (n=728 976).

*Weighted percentages are calculated using survey design and non-response weights (by age, gender, geographical location and GP practice, full details Technical Annex[13]).

†OR prefer nurse come from the same model as prefer GP and are equal to 1/(OR prefer GP).

‡P<0.001 for all except gender where P=0.969.

§P<0. 001 for all except gender where P=0.024.

GP, general practitioner.

extends on smaller qualitative studies of patients' preferences for GP versus nurse-led consultations,[10 11] controlled trials investigating the impacts of nurse-led primary care on hospital admission and mortality,[18] and qualitative synthesis of evidence on the barriers and facilitators to the implementation of doctor–nurse substitution strategies in primary care.[19] In particular, this study highlights the possible unintended impacts of nurse–GP substitution on patients' confidence and trust, and the quality of communication in primary care. The response rate (34%) in our study is comparable with other major national surveys, and a previous assessment found no association between

**Table 3** Results of the regression analyses examining the association between patient-rated confidence and trust in health professionals, and the difference in the type of appointment patients wanted and the appointment they got at their last consultation in primary care

| Patient request | Appointment outcome | N (%) | Patient-rated confidence and trust | Crude difference* (95% CI) | Adjusted difference† (95% CI) |
|---|---|---|---|---|---|
| Consultation outcome: saw/spoke to nurse | | | | | |
| See a nurse | Saw a nurse | 105 713 (87.0) | 91.0 | Reference | Reference |
| See a nurse | Spoke to a nurse | 518 (0.4) | 87.6 | −3.40 (−5.27 to −1.53) | −2.87 (−4.70 to −1.04) |
| Speak to a nurse | Saw a nurse | 1165 (1.0) | 87.8 | −3.23 (−4.48 to −1.98) | −2.82 (−4.05 to −1.60) |
| Speak to a nurse | Spoke to a nurse | 1698 (1.4) | 88.3 | −2.70 (−3.74 to −1.66) | −2.40 (−3.42 to −1.38) |
| See a GP | Saw a nurse | 10 769 (8.9) | 79.5 | −11.58 (−12.01 to −11.15) | −9.90 (−10.33 to −9.47) |
| See a GP | Spoke to a nurse | 534 (0.4) | 72.9 | −18.10 (−19.94 to −16.26) | −15.83 (−17.64 to −14.02) |
| Speak to a GP | Saw a nurse | 821 (0.7) | 83.9 | −7.18 (−8.67 to −5.69) | −6.04 (−7.50 to −4.58) |
| Speak to a GP | Spoke to a nurse | 286 (0.2) | 76.0 | −14.99 (−17.51 to −12.48) | −13.47 (−15.93 to −11.01) |
| Consultation outcome: saw/spoke to GP | | | | | |
| See a GP | Saw a GP | 446 631 (90.6) | 84.4 | Reference | Reference |
| See a GP | Spoke to a GP | 9000 (1.8) | 76.7 | −7.68 (−8.22 to −7.13) | −6.47 (−7.00 to −5.93) |
| Speak to a GP | Saw a GP | 8190 (1.7) | 83.6 | −0.82 (−1.40 to −0.25) | −0.77 (−1.33 to −0.22) |
| Speak to a GP | Spoke to a GP | 21 959 (4.5) | 86.4 | 1.96 (1.61 to 2.32) | 1.42 (1.07 to 1.76) |
| See a nurse | Saw a GP | 5763 (1.2) | 84.2 | −0.19 (−0.87 to 0.49) | −0.73 (−1.39 to −0.08) |
| See a nurse | Spoke to a GP | 370 (0.1) | 79.2 | −5.21 (−7.88 to −2.54) | −5.63 (−8.20 to −3.07) |
| Speak to a nurse | Saw a GP | 893 (0.2) | 82.9 | −1.48 (−3.20 to 0.24 | −1.48 (−3.13 to 0.18) |
| Speak to a nurse | Spoke to a GP | 173 (0.1) | 80.6 | −3.76 (−7.67 to 0.14) | −4.04 (−7.80 to −0.29) |

*P<0.001 (joint test).
†Also adjusted for age, gender, ethnicity, confidence in managing their own health, the presence of a long-standing heath condition, deprivation (fixed effects) and practice (random effect) P<0.001 (joint test).
GP, general practitioner.

practice response rates and scores.[20] Despite its large overall sample, one limitation of our study is the smaller size of some individual patient subgroups giving rise to larger uncertainty. Another limitation is that we assessed patient-reported confidence and trust using the single item that represents this concept in the GPPS, and future research might consider including additional items to provide a comprehensive and multidimensional measure of these constructs.

### Changing nature of primary care and implications for patient confidence and trust

The role of nurses in primary care has changed considerably in the last 30 years, with the advent of advanced nurse practitioners, independent nurse prescriber roles and the existence of some primary care practices that are led by nurses. Some patients appear uncertain about what to expect in nurse-led primary care consultations.[9 11] In our study, we found that patients who got to speak to a nurse when they wished to see a GP, reported lowest confidence and trust in their eventual consultation. The magnitude of this effect represents a large difference of 15 points on a scale of 0–100, controlling for sociodemographic characteristics and health status and combines not seeing or speaking to the type of practitioner of their choice (ie, nurse rather than doctor) and receiving a telephone

call when they had asked for a face-to-face consultation. There are several possible explanations for these findings. Patients may feel that they are being inappropriately denied access to GPs, resulting in frustration that could affect both trust and the quality of interpersonal communication (gatekeeper hypothesis). Lack of trust and confidence may also arise because some patients perceive that nurses are subordinate to GPs in terms of skills and knowledge (patient confidence hypothesis). Our findings are in line with the results from a pragmatic controlled trial of telephone triage among patients seeking a same-day GP appointment which showed that nurse-led triage was associated with somewhat worse patient-reported experience and lower overall patient satisfaction in comparison with usual care.[21]

### Implications for health policy, clinical practice and primary care research

A strong vision for reforming primary care in England is now emerging.[6] This includes expanded roles for pharmacists, physician associates and a £15 million investment in nursing capacity within general practice. When new ways of working are introduced, primary care practices need to communicate these to patients who may know less about the care offered by health professionals other than GPs. Building patient confidence and educating patients

**Table 4** Results of regression analyses examining the association between patient-rated nurse and GP communication scores, and the difference in the type of appointment patients wanted and the appointment they got at their last consultation in primary care

| Patient request | Appointment outcome | N (%) | Mean Communication Score | Crude difference* (95% CI) | Adjusted difference† (95% CI) |
|---|---|---|---|---|---|
| **Consultation outcome: saw/spoke to nurse** | | | | | |
| See a nurse | Saw a nurse | 105 140 (86.8) | 90.0 | Reference | Reference |
| See a nurse | Spoke to a nurse | 517 (0.4) | 88.3 | −1.70 (−2.97 to −0.44) | −1.40 (−2.63 to −0.18) |
| Speak to a nurse | Saw a nurse | 1170 (1.0) | 87.6 | −2.37 (−3.21 to −1.53) | −2.05 (−2.87 to −1.23) |
| Speak to a nurse | Spoke to a nurse | 1697 (1.4) | 88.9 | −1.09 (−1.79 to −0.38) | −0.93 (−1.61 to −0.24) |
| See a GP | Saw a nurse | 10916 (9.0) | 82.7 | −7.31 (−7.60 to −7.02) | −5.99 (−6.28 to −5.71) |
| See a GP | Spoke to a nurse | 538 (0.4) | 77.8 | −12.16 (−13.40 to −10.92) | −10.47 (−11.68 to −9.27) |
| Speak to a GP | Saw a nurse | 819 (0.7) | 85.7 | −4.25 (−5.26 to −3.25) | −3.35 (−4.33 to −2.38) |
| Speak to a GP | Spoke to a nurse | 289 (0.2) | 80.2 | −9.80 (−11.49 to −8.11) | −8.66 (−10.30 to −7.03) |
| **Consultation outcome: saw/spoke to GP** | | | | | |
| See a GP | Saw a GP | 450 555 (90.6) | 85.4 | Reference | Reference |
| See a GP | Spoke to a GP | 9127 (1.8) | 80.0 | −5.41 (−5.77 to −5.05) | −4.49 (−4.84 to −4.14) |
| Speak to a GP | Saw a GP | 8281 (1.7) | 85.2 | −0.21 (−0.59 to 0.17) | −0.23 (−0.59 to 0.13) |
| Speak to a GP | Spoke to a GP | 22 039 (4.4) | 86.9 | 1.54 (1.30 to 1.77) | 0.96 (0.73 to 1.19) |
| See a nurse | Saw a GP | 5831 (1.2) | 84.5 | −0.85 (−1.30 to −0.40) | −1.22 (−1.65 to −0.80) |
| See a nurse | Spoke to a GP | 378 (0.1) | 82.0 | −3.41 (−5.17 to −1.64) | −3.76 (−5.43 to −2.10) |
| Speak to a nurse | Saw a GP | 913 (0.2) | 84.3 | −1.07 (−2.21 to 0.07) | −0.94 (−2.01 to 0.13) |
| Speak to a nurse | Spoke to a GP | 178 (0) | 83.7 | −1.64 (−4.21 to 0.93) | −2.16 (−4.59 to 0.26) |

*P<0.001 (joint test).
†Also adjusted for age, gender, ethnicity, confidence in managing their own health, the presence of a long-standing heath condition, deprivation (fixed effects) and practice (random effect) P<0.001 (joint test).
GP, general practitioner.

about the skills of team members should form part of a broader effort to involve patients in the design and development of new services. Involving patients in service redesign may help to address barriers that affect patient willingness to consult with nurses or physician associates, for example, patient-reported experiences of incomplete or delayed care (including prescription delay), and concerns about the loss of provider continuity.[22]

In developing new models of working, staff training needs to be carefully considered. A study of nurse-led clinics for patients with osteoarthritis in general practice found a significant gap between National Institute for Health and Care Excellence recommendations and the care nurses felt confident and able to deliver.[23] Staff transitioning in to new roles in patient care or in patient triage need to feel adequately prepared, and standardised training may be necessary, but insufficient, to ensure success.[24] New models of care should consider the possible intended and unintended consequences for patient experience of multidisciplinary teams, particularly given the policy emphasis on enabling patient choice. Monitoring should include any potential adverse effects on patient confidence and trust in health professionals. Qualitative interviews with patients' consulting a physician associate in general practice highlight a desire for continuity with a trusted clinician, the importance of patient choice,

and the maintenance of trust and confidence in general practice.[22]

In the context of ageing populations, general practices are seeing increasing numbers of patients with complex health needs and multiple long-term conditions.[25] New models of care in England emerging, for example, through the multispecialty community provider contract framework,[26] need to consider how to make the best use of the primary care workforce. Many GP practices now offer nurse-led consultations for patients with chronic conditions, which are often age-related, such as diabetes. Our results provide some support for this model of care, with approximately one in five patients aged over 65 wanting an appointment with a nurse. Among patients who wanted a nurse appointment and got one, patient-reported trust and confidence in the nurse consultation was very high (mean score 91.0 on a scale of 0–100). Further work is needed to ascertain the impacts of proposed workforce changes on different patient groups. While proposals for nurses to help meet gaps in the supply of primary care providers have met with wide interest,[27] there are important remaining uncertainties about the impact of this on patients' experiences and on the maintenance of confidence and trust in primary care in the NHS.

**Contributors** All authors (CAMP, GAA, JB, JLC, MNE, VAL and MR) jointly conceived of the research questions, reviewed/edited the manuscript and contributed to discussion and interpretation of the data. CAMP and GAA led on writing the manuscript and GAA completed the statistical analysis. All authors (CAMP, GAA, JB, JLC, MNE, VAL and MR) had full access to all of the data in the study and take responsibility for the integrity of the data and accuracy of the data analysis. CAMP is the guarantor of the paper.

**Funding** This paper presents independent research funded by the National Institute for Health Research (NIHR) under its Programme Grants for Applied Research Programme (Grant Reference Number RP-PG-0608-10050). The study was sponsored by the University of Cambridge.

**Disclaimer** The views expressed are those of the author(s) and not necessarily those of the NHS, the NIHR or the Department of Health. The researchers confirm their independence from the study funders, the National Institute for Health Research.

**Competing interests** None declared.

**Patient consent** Not required.

**Provenance and peer review** Not commissioned; externally peer reviewed.

**Data sharing statement** No additional data are available.

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
