## [Reviewer comments · BMJ Open]

ARTICLE DETAILS

TITLE (PROVISIONAL)	What happens to patient experience when you want to see a doctor and you get to speak to a nurse? Observational study using data from the English General Practice Patient Survey
AUTHORS	Paddison, Charlotte; Abel, Gary; Burt, Jenni; Campbell, John; Elliot, Marc; Lattimer, Valerie; Roland, Martin

VERSION 1 – REVIEW

REVIEWER	Ludmila Marcinowicz Medical University of Bialystok, Poland
REVIEW RETURNED	17-Aug-2017

GENERAL COMMENTS	This is an interesting study and adds to the body of literature on patient preferences for seeing or speaking to a GP or a nurse. The paper is well structured and easy to read. The methods are appropriate, executed well and the results are presented well and clearly. The discussion contains information that adds value and is interesting to read. Thank you very much for the opportunity to read your study.
---

REVIEWER	Susan Waterworth Senior Lecturer, Faculty of Medical and Health Sciences, The University of Auckland, New Zealand
REVIEW RETURNED	18-Oct-2017

GENERAL COMMENTS	Thank you and this was an interesting manuscript to review. The issue of expanding multidisciplinary teams in General practice has been discussed for some time now and certainly the value of the nursing role. Your findings certainly present the different perspectives of how patients' view the GP/Nurse and how this partnership could be developed further to harness the contribution of each to maximise the patient experience. You identified the responsibility of the General practice to make clear what services the GP/Nurse can provide. There are more questions and these are the areas that you could expand in your discussion. For example, the patients who did not feel confident in managing their health were wanting to see the GP, yet the ones that were confident were satisfied in seeing the nurse.
---

	You have provided some hypotheses for these; gatekeeper hypothesis and patient confidence hypothesis (any references?) Importantly with the projected increase in the ageing population, your sample of older patients were satisfied with seeing a nurse, so any further discussion of this finding would be warranted. Your findings reflect the need to involve patients in designing service changes that can support them in managing their own health, and I agree, but is there any evidence that this approach would resolve the issues you have identified? There has been a lot of work on co-design in health and I wondered if any of this would enable you to develop the discussion further. The need to consider the intended and unintended consequences of new models of care is an important point. You have identified the limitations of your study and the need for further research. In particular the single item reporting trust and confidence. The question on whether the patient is involved in making decisions about their care was not reported and I wondered why that was the case? I can see how your focus was more on the 'who they wanted to see'. On page 4. line 13 you appear to suggest that one of the reasons for the expanded workforce is to free up GP time, particularly on minor issues. Importantly, there is evidence that the patients being seen by the other members of the team do not necessarily manage patients with minor health issues. In fact, the evidence is that nurses are managing patients with complex co-morbidities. Some of your references are rather dated; 8, 9, 10 and 11 and there are more recent papers available. There is no date on the Campbell et al reference.
--	--

REVIEWER	Jane Desborough Australian National University, Australia
REVIEW RETURNED	26-Oct-2017

GENERAL COMMENTS	Thank you for the opportunity to review this interesting and very relevant article. The results are important in terms of understanding how to manage and implement interprofessional teams in primary care. I have only one comment to make and that is that while the biggest differences in trust & confidence, and communication ratings were found in patients who wanted to see a GP and got to speak to a nurse, it is clear that there was a significant difference (although not as great) for patients who wanted to see a nurse and got to speak to a GP. The discussion focuses on the first of these, although I think it is important to refer to the latter briefly. In particular, it is clear that relationships are important and for some, those with nurses are equally as important as with the GP's. Older people were more likely to ask to see a nurse, perhaps due to the qualities that previous research has indicated that patients value in nurses - having more time to talk, and listening ... (Phillips et al 2007, Laurent 2005, Desborough, 2017, Halcomb et al., 2013; Rohrer, Wilshusen, Adamson, & Merry, 2008)
---

VERSION 1 – AUTHOR RESPONSE

Reviewer: 1

Reviewer Name: Ludmila Marcinowicz

Institution and Country: Medical University of Bialystok, Poland Competing Interests: None declared.

Comment: This is an interesting study and adds to the body of literature on patient preferences for seeing or speaking to a GP or a nurse. The paper is well structured and easy to read. The methods are appropriate, executed well and the results are presented well and clearly.

The discussion contains information that adds value and is interesting to read. Thank you very much for the opportunity to read your study.

Authors' response: We would like to thank Dr Marcinowicz for her time in reviewing our manuscript and providing comments.

Reviewer: 2

Reviewer Name: Susan Waterworth

Institution and Country: Senior Lecturer, Faculty of Medical and Health Sciences, The University of Auckland, New Zealand

Comment: Thank you and this was an interesting manuscript to review. The issue of expanding multidisciplinary teams in General practice has been discussed for some time now and certainly the value of the nursing role. Your findings certainly present the different perspectives of how patients' view the GP/Nurse and how this partnership could be developed further to harness the contribution of each to maximise the patient experience. You identified the responsibility of the General practice to make clear what services the GP/Nurse can provide. There are more questions and these are the areas that you could expand in your discussion. For example, the patients who did not feel confident in managing their health were wanting to see the GP, yet the ones that were confident were satisfied in seeing the nurse. You have provided some hypotheses for these; gatekeeper hypothesis and patient confidence hypothesis (any references?) Importantly with the projected increase in the ageing population, your sample of older patients were satisfied with seeing a nurse, so any further discussion of this finding would be warranted. Your findings reflect the need to involve patients in designing service changes that can support them in managing their own health, and I agree, but is there any evidence that this approach would resolve the issues you have identified? There has been a lot of work on co-design in health and I wondered if any of this would enable you to develop the discussion further.

The need to consider the intended and unintended consequences of new models of care is an important point.

You have identified the limitations of your study and the need for further research. In particular the single item reporting trust and confidence. The question on whether the patient is involved in making decisions about their care was not reported and I wondered why that was the case? I can see how your focus was more on the 'who they wanted to see'.

On page 4. line 13 you appear to suggest that one of the reasons for the expanded workforce is to free up GP time, particularly on minor issues. Importantly, there is evidence that the patients being seen by the other members of the team do not necessarily manage patients with minor health issues. In fact, the evidence is that nurses are managing patients with complex co-morbidities. Some of your references are rather dated; 8, 9, 10 and 11 and there are more recent papers available. There is no date on the Campbell et al reference.

Authors response:

We welcome the suggestions from Reviewer 2, and in particular her ideas on where we might further expand in the 'discussion' section.

In responding to her suggestions we have added further text on page 8 as follows:

“In the context of aging populations, general practices are seeing increasing numbers of patients with complex health needs, and multiple long-term conditions. New models of care in England emerging through the multispecialty community provider (MCP) contract framework, for example, need to consider the how to make best use of the primary care workforce. Many GP practices now offer nurse-led consultations for patients with chronic conditions which are often age-related, such as diabetes. Our results provide some support for this model of care, with approximately one in five patients aged over 65 wanting an appointment with a nurse. Among patients who wanted a nurse appointment and got one, patient-reported trust and confidence in the nurse consultation was very high (mean score 91.0 on a scale of 0 -100).”

Although Reviewer 2 observed from our study that older patients were more satisfied with seeing a nurse and the implications of this for providing care for an aging population, we did not measure or report patient satisfaction in this study, and nor did we in this paper provide a breakdown of patient-reported trust and confidence in nurses by age group. Therefore we would be cautious about claiming that older patients were more satisfied with seeing a nurse. Instead, we have noted on page 8 that patients (of all ages) who wanted a nurse appointment and got one had high patient-reported trust and confidence scores, and, separately, that older adults were more likely to want an appointment with a nurse (possibly due to the provision of nurse-led appointments in chronic disease clinics, which are well-attended by older adults).

We have reviewed literature as Reviewer 2 suggested on co-design in health, and we did consider expanding our discussion to include literature on the benefits of involving patients more in designing service changes that can help them in managing their own health, as Reviewer 2 suggested. On balance, we did not feel that this change would strengthen the discussion enough to warrant inclusion. Our main point was slightly different. We focused on page 8 on the importance of building patient confidence and educating patients about the skills of different primary care professionals (e.g., GP, nurses, physicians assistants), which we see as part of broader efforts to involve patients in the design and development of new services. This discussion wasn't focused on building patient confidence or education for self-care, which we see as a somewhat different issue. We did however add the following text on page 8, paragraph two, which further emphasises the importance of involving patients in service redesign to help address areas of poor patient experience already identified in the literature:

“Involving patients in service redesign may help to address barriers that affect patient willingness to consult with nurses or physician associates, for example patient-reported experiences of incomplete or delayed care (including prescription delay), and concerns about the loss of provider continuity.”

We do not report in the paper on patient responses to the survey question that asks whether the patient was involved in decisions about their care, as this is beyond the scope of this paper. We focus in this paper on addressing the two main research questions on page 4, which are:

1. To what extent do patients in primary care request an appointment with a doctor or a nurse, and how often is this request met?
2. When patients are unable to get an appointment of their choice (phone or face-to-face, GP or nurse), is this associated with poorer patient-reported communication or lower trust and confidence?

On page 4 our first paragraph gives a broad overview of the policy and clinical context for this research. In this paragraph, we suggest that one of the reasons for the expanded workforce is to free up GP time, enabling them to devote more time to the delivery of care for complex patients and spend less time on administration and minor health issues – this is key line of argument in recent UK policy within primary care (e.g., Roland et al (2015), The future of primary care, Creating teams for

tomorrow). We also acknowledge that other members of the team might manage patients with both minor health issues, and those with complex issues (who now make up a large proportion of primary care consultations). We perceive that both statements can be true: (a) It is anticipated that an expanded workforce will free up clinical time for doctors, enabling them to devote more time to the delivery of care for complex patients and spend less time on administration and minor health issues (which is what we say), and (b) other members of the team do not necessarily manage patients only with minor health issues, for example nurses also manage patients with complex co-morbidities (as Reviewer 2 observes). The reasons given for the expanded workforce in recent UK policy documents are as we have described them on page 4 (enabling GPs to devote more time to the delivery of care for complex patients and spend less time on administration and minor health issues - see 'The future of primary care, Creating teams for tomorrow'

<https://www.hee.nhs.uk/sites/default/files/documents/The%20Future%20of%20Primary%20Care%20Report.pdf>) and we have added a reference to page 4 line to reflect this link to policy more clearly.

We have added additional references, to include relevant and recent literature. These are detailed at the end of this document.

We have added a date on the Campbell et al reference. Thank you for pointing this out to us.

The authors would like to thank Dr Waterworth for her time in reviewing our manuscript and her constructive comments and help in strengthening the manuscript.

Reviewer: 3

Reviewer Name: Jane Desborough

Institution and Country: Australian National University, Australia Competing Interests: None declared

Thank you for the opportunity to review this interesting and very relevant article. The results are important in terms of understanding how to manage and implement interprofessional teams in primary care. I have only one comment to make and that is that while the biggest differences in trust & confidence, and communication ratings were found in patients who wanted to see a GP and got to speak to a nurse, it is clear that there was a significant difference (although not as great) for patients who wanted to see a nurse and got to speak to a GP. The discussion focuses on the first of these, although I think it is important to refer to the latter briefly. In particular, it is clear that relationships are important and for some, those with nurses are equally as important as with the GP's. Older people were more likely to ask to see a nurse, perhaps due to the qualities that previous research has indicated that patients value in nurses - having more time to talk, and listening ... (Phillips et al 2007, Laurent 2005, Desborough, 2017, Halcomb et al., 2013; Rohrer, Wilshusen, Adamson, & Merry, 2008)

Authors response:

We welcome the suggestions from Reviewer 3, and thank her for her time in reviewing our manuscript.

We agree that there was a significant difference in patient-related experience for those who wanted to see a nurse and got to speak to a GP, and we have added text to the discussion on page 7, paragraph 2, to reflect this.

"A difference, though smaller in magnitude, was also observed for patients who wanted to see a nurse and got to speak to a GP."

We also agree that relationships with nurses are important. In response to comments from both reviewers 2 and 3 we have added an additional paragraph on page 8, on the implications of our research findings for models of care in the context of aging populations, specifically highlighting the important contribution of nurse-led consultations for chronic illness. In this expanded discussion we now note that "approximately one in five patients aged over 65 wanted an appointment with a nurse,

and that, among patients (of all ages) who wanted a nurse appointment and got one, patient-reported trust and confidence in the nurse consultation was very high (mean score 91.0 on a scale of 0 -100).”

New references added (to include relevant and recent literature as requested by reviewer 2):

Varley A, Warren FC, Richards SH, et al. The effect of nurses' preparedness and nurse practitioner status on triage call management in primary care: A secondary analysis of cross-sectional data from the ESTEEM trial. *International Journal of Nursing Studies*. 2016;58:12-20.
doi:10.1016/j.ijnurstu.2016.02.001.

Healey EL, Main CJ, Ryan S, et al. A nurse-led clinic for patients consulting with osteoarthritis in general practice: development and impact of training in a cluster randomised controlled trial. *BMC Family Practice*

Rashidian A, Shakibazadeh E, Karimi- Shahanjarini A, et al. Barriers and facilitators to the implementation of doctor-nurse substitution strategies in primary care: qualitative evidence synthesis. In: *Cochrane Database of Systematic Reviews*. 2013. doi:10.1002/14651858.CD010412

Halcomb EJ, Peters K, Davies D. A qualitative evaluation of New Zealand consumers perceptions of general practice nurses. *BMC Fam Pract* 2013;14:26. doi:10.1186/1471-2296-14-26

Martínez-González NA, Djalali S, Tandjung R, et al. Substitution of physicians by nurses in primary care: a systematic review and meta-analysis. *BMC Health Serv Res* 2014;14:214.

Parker R, Forrest L, Ward N, et al. How acceptable are primary health care nurse practitioners to Australian consumers? *Coll R Coll Nurs Aust* 2013;20:35–41.

Halter M, Drennan VM, Joly LM, et al. Patients' experiences of consultations with physician associates in primary care in England: A qualitative study. *Health Expect* 2017;20:1011–9. doi:10.1111/hex.12542

Barnett K, Mercer SW, Norbury M, et al. Epidemiology of multimorbidity and implications for health care, research, and medical education: a cross-sectional study. *The Lancet* 2012;380:37–43.

NHS England. Multispeciality community (MCP) emerging care model and contract framework. 2016. <https://www.england.nhs.uk/wp-content/uploads/2016/07/mcp-care-model-frmrwrk.pdf>

VERSION 2 – REVIEW

REVIEWER	Jane Desborough Research School of Population Health, Australian National University
REVIEW RETURNED	16-Dec-2017
GENERAL COMMENTS	Thank you for the opportunity to review this manuscript again and for the thorough way in which you have addressed the reviewers' comments. I believe it meets the requirements for publication. Congratulations.